# Antagonistic Activity of Oroxylin A against *Fusarium graminearum* and Its Inhibitory Effect on Zearalenone Production

**DOI:** 10.3390/toxins15090535

**Published:** 2023-08-31

**Authors:** Luli Zhou, Guanyu Hou, Hanlin Zhou, Khaled Abouelezz, Yuxiu Ye, Jun Rao, Song Guan, Dingfa Wang

**Affiliations:** 1Tropical Crops Genetic Resources Institute, Chinese Academy of Tropical Agricultural Sciences, Haikou 571101, China; zhoull@catas.cn (L.Z.); guanyuhou@126.com (G.H.); guansong79@126.com (S.G.); 2Zhanjiang Experimental Station, Chinese Academy of Tropical Agricultural Sciences, Zhanjiang 524013, China; zhouhanlin8@163.com; 3Department of Poultry Production, Faculty of Agriculture, Assiut University, Assiut 71526, Egypt; abollez@aun.edu.eg; 4Hainan Yitian Biotechnology Co., Ltd., Haikou 570228, China; 13322059270@163.com; 5Tianjin Ninghe Original Pig Farm Co., Ltd., Tianjin 301500, China; raojun_j@163.com

**Keywords:** *Piper sarmentosum* extract, Oroxylin A, *Fusarium graminearum*, zearalenone, metabolic pathways

## Abstract

*Fusarium graminearum* produces zearalenone (ZEA), a mycotoxin that is widely found in food and feed products and is toxic to humans and livestock. *Piper sarmentosum* extract (PSE) inhibits *F. graminearum*, and Oroxylin A appears to be a major antifungal compound in PSE. The aim of this study is to quantify the Oroxylin A content in PSE using UPLC-QTOF-MS/MS, and to investigate the antagonistic activity of Oroxylin A against *F. graminearum* and its inhibitory effect on ZEA production. The results indicate that Oroxylin A inhibits both fungal growth and ZEA production in a dose-dependent manner. Oroxylin A treatment downregulated the mRNA expression of zearalenone biosynthesis protein 1 (*ZEB1*) and zearalenone biosynthesis protein 2 (*ZEB2*). The metabolomics analysis of *F. graminearum* mycelia indicated that the level of ribose 5-phosphate (R5P) deceased (*p* < 0.05) after Oroxylin A treatment (64–128 ng/mL). Moreover, as the Oroxylin A treatment content increased from 64 to 128 ng/mL, the levels of *cis*-aconitate (*p* < 0.05) and fumarate (*p* < 0.01) were upregulated successively. A correlation analysis further showed that the decreased R5P level was positively correlated with *ZEB1* and *ZEB2* expression, while the increased *cis*-aconitate and fumarate levels were negatively correlated with *ZEB1* and *ZEB2* expression. These findings demonstrate the potential of Oroxylin A as a natural agent to control toxigenic fungi and their mycotoxin.

## 1. Introduction

Zearalenone (ZEA) is a mycotoxin that is mainly produced by Fusarium species, including *F. graminearum*, which is found to contaminate a wide range of foods and feeds [1,2]. ZEA has a high thermal stability, so it is difficult to degrade during the processing, storage, or handling of foods and feeds, constituting a considerable threat to human and animal health [3,4]. After ingestion, ZEA and its reduced metabolites can bind to and activate estrogen receptors because of their structural similarity to estrogens, which, in turn, causes reproductive dysfunction in female animals such as abortion and false estrus, as well as immunotoxicity and advanced puberty [5].

In recent years, many synthetic antifungal agents have been applied to inhibit the growth of *F. graminearum* and other fungi, but the use of synthetic antifungal agents has resulted in adverse effects on the environment, the safety of agricultural products, and human and livestock health [6]. Therefore, there is a need to find new and less harmful antifungal and mycotoxin controlling agents; fortunately, multiple active ingredients in plant extracts provide excellent options as alternatives to synthetic antifungal agents due to their good antifungal activities and bio-degrading nature and because they are not toxic to the environment [7,8,9,10].

*P. sarmentosum* is an edible herb from the *Piperaceae* (pepper) family; it is widely distributed throughout tropical and sub-tropical regions of the world, and it is commonly used in cooking. It has antioxidant [11], anti-inflammatory [12], and antimicrobial activities [13], and these beneficial health effects have been primarily associated with alkaloids and phenolic compounds [12,14,15]. In our previous study, *P. sarmentosum* extract (PSE) exerted antifungal activity against *F. graminearum* at doses of 1 and 2 mg/mL, and 17 compounds were tentatively identified as the main antifungal constituents of PSE [16]. Among the 17 compounds, Oroxylin A was the only flavonoid, and this was the first report of Oroxylin A in *P. sarmentosum*. Studies have shown that Oroxylin A has a broad spectrum of pharmacological effects including anti-inflammatory, anti-cancer, and anti-bacterial effects, and so on [17,18]. Meanwhile, it was suggested that Oroxylin A could be functional in multiple signaling pathways and could be a promising and effective compound in the development of new additives, nutraceuticals, and therapeutic agents [18,19]. However, the antifungal activity of Oroxylin A against *F. graminearum* has not been extensively studied.

In the past, we merely performed the tentative identification of Oroxylin A in PSE through the mzCloud database (https://www.mzcloud.org, accessed on 30 May 2021) [16]. Therefore, in the present study, UPLC-QTOF-MS was used for further qualitative and quantitative validation of Oroxylin A in PSE based on the MS and MS^2^ fragmentation patterns of an Oroxylin A reference standard in order to enable the determination of the suitable dosages of Oroxylin A for subsequent experiments. In addition, the antifungal and anti-mycotoxin properties of Oroxylin A against *F. graminearum* were assessed.

## 2. Results

### 2.1. Qualitative and Quantitative Analyses of Oroxylin A in Piper Sarmentosum Extract (PSE)

Oroxylin A was characterized in PSE using UPLC-QTOF-MS/MS. The extracted ion chromatogram (EIC) and MS/MS fragmentation patterns of Oroxylin A in PSE are shown in Figure 1A,B, respectively. Oroxylin A (C_16_H_12_O_5_) had the following characteristics: retention time (Rt), 3.11 min; [M-H]^−^/*m*/*z*, observed value of 283.0351, calculated value of 283.0606, and mass error of −9.0 ppm; and MS^2^ fragment ion, 268.0136 [M-H-CH_3_]^−^. The identification of the observed compound as Oroxylin A was confirmed (Figure 1B) by the [M-H]^−^ precursor ion at *m*/*z* 283.0351 [C_16_H_12_O_5_] and CID of [M-H]^−^ (loss of CH_3_; 15 Da) to generate a fragment ion at *m*/*z* 268.0136 [C_15_H_9_O_5_], which is consistent with our previous MS-based qualitative analysis of Oroxylin A (Appendix A) as well as the comparison with standard Oroxylin A in this study.

A calibration graph was plotted based on a linear regression analysis of the integrated peak areas (y) versus the concentrations (x, ng/mL) of Oroxylin A in the standard solution (regression equation, y = 571.80x − 10575.42; linear range, 20–500 ng/mL; R^2^ = 0.9937), which was used to determine the Oroxylin A content of PSE as 64 ng/mg in PSE.

A previous study [16] found that 1 and 2 mg/mL of PSE had antifungal activities against *F. graminearum* and that Oroxylin A is one of the main antifungal compounds of PSE. In this study, 1 and 2 mg/mL of PSE were found to contain 64 ng and 128 ng Oroxylin A, respectively; therefore, the three test concentrations of Oroxylin A used here were 0 (T0; control), 64 (T1), and 128 ng/mL (T2).

### 2.2. Antifungal Activity of Oroxylin A against F. graminearum

Oroxylin A contents of 64 and 128 ng/mL both inhibited the growth of *F. graminearum* on agar plates 3 and 5 days after inoculation (Figure 2A,B), and the inhibition was quantified (Figure 2C). An Oroxylin A content of 64 ng/mL (T1) inhibited growth by 22.4 and 15.0% after 3 and 5 days of incubation, respectively, and 128 ng/mL (T2) of Oroxylin A inhibited growth by 43.4 and 20.1% after 3 and 5 days, respectively. The T2 inhibition was higher than that of T1 (*p* < 0.05).

### 2.3. Anti-Mycotoxin Activity of Oroxylin A against F. graminearum

The anti-mycotoxin activity of Oroxylin A against *F. graminearum* was determined; there was a downward trend in the ZEA production with an increasing Oroxylin A concentration (Figure 3A), and an Oroxylin A content of 128 ng/mL significantly inhibited ZEA production compared with T0 (*p* < 0.05).

To help elucidate the mechanism of ZEA production inhibition by Oroxylin A, the mRNA expressions of two key genes (*ZEB1* and *ZEB2*) that regulate ZEA synthesis were assessed. Oroxylin A at both treatment concentrations markedly and similarly reduced *ZEB1* expression (*p* < 0.001) (Figure 3B), whereas it reduced *ZEB2* expression less strongly, but in a dose-dependent manner (*p* < 0.01).

### 2.4. Untargeted Metabolomic Analysis of Oroxylin A-Treated F. graminearum

The global metabolomic profiling of *F. graminearum* mycelia detected 201 metabolite features across all experimental groups. To examine the metabolic differences in *F. graminearum* resulting from the T0, T1, and T2 treatments, the metabolomic data were analyzed using unsupervised PCA; the score plot revealed that PC1 accounted for 23.7% of the total variation and PC2 accounted for 14.3% of the total variation (Figure 4A). However, the separation between the three treatments was poor, and this model was unable to clearly distinguish between the treatments.

Supervised PLS-DA was applied to the data (Figure 4B), which resulted in a clear separation between the groups and a marked dose–response relationship, with the T0/T2 separation being approximately double that of the T0/T1 separation. This also indicates a dose–response relationship in the metabolic changes induced by Oroxylin A treatment.

Among all the metabolites detected in the fungal mycelium, there were nine differential metabolites between T1 and T0, and eight differential metabolites between T2 and T0 (Appendix A). Out of these, cis-aconitate and fumarate are two important intermediate metabolites of the TCA cycle. It was noticed that compared with T0, the cis-aconitate concentration (*p* < 0.05) in T1 and that of fumarate (*p* < 0.01) in T2 both increased (Figure 5A). In addition, the concentration of ribose-5-phosphate (R5P) decreased in both T1 and T2 (*p* < 0.05); this was the only differential metabolite that was common to both treatments (Figure 5B).

### 2.5. Correlation Analysis between Differential Metabolites and Expression Changes of ZEB1 and ZEB2 Genes

Pearson’s correlation analysis (Figure 6) showed significant correlations between most of the differential metabolites and the *ZEB1* and/or *ZEB2* mRNA levels. Seven metabolites correlated with both *ZEB1* and *ZEB2* (*p* < 0.05), namely aminoadipic acid, *cis*-aconitate, p-aminobenzoate, and R5P in T1 (Figure 6A), and spermine, fumarate, and *L*-homocysteic acid in T2 (Figure 6B).

## 3. Discussion

In our previous study, we found that *P. sarmentosum* extract (PSE) inhibited *F. graminearum* growth at concentrations of 1 and 2 mg/mL, and Oroxylin A was tentatively identified as a major contributor to *F. graminearum* inhibition in PSE [16]. In this study, the Oroxylin A of the PSE was further qualified and quantified based on UPLC-QTOF-MS/MS analysis by using the authentic standard. The analyses revealed a very low Oroxylin A content of 64 ng/mg in PSE. Subsequently, the in vitro antifungal activity evaluation demonstrated that Oroxylin A showed effective inhibitory activity against *F. graminearum* at concentrations of both 64 and 128 ng/mL, and the inhibition was dose-dependent. Moreover, we also found that pure Oroxylin A treatment inhibited ZEA production and downregulated the expressions of *ZEB1* and *ZEB2*. It was known that ZEA is a polyketide mycotoxin produced by *F. graminearum* and its biosynthesis is closely related to the expressions of two genes (*ZEB1* and *ZEB2*) [20]. This may suggest that *ZEB1* and *ZEB2* act as effector sites for Oroxylin A and are involved in regulating ZEA production.

Previous work has revealed that Oroxylin A is a methoxyflavone, belongs to the class of phenolic compounds, and displays diverse biological activities including antioxidant, anti-inflammatory, antimicrobial, and immunomodulatory activities [21], while the antimicrobial activities of phenolic compounds result from their hydrophobic characteristic. And this chemical characteristic permits them to cross the lipid layers of the cell membrane and interact with cell compounds, which could lead to a deformation in the cell structure and functionality, ultimately causing cell death [6]. It was also reported that the phenolic hydroxyl group at the C-7 position of Oroxylin A is essential for antimicrobial activity [22], and the presence of a methoxyl (-OCH_3_) group at C-6 contributes to its ability to bind to intracellular membrane structures, which may improve its bioavailability and antimicrobial activity [23,24]. It appears that the inhibitory effects of Oroxylin A against *F. graminearum* are related to its chemical structure.

Metabolically, there are some similarities and differences of the differential metabolites in the groups treated with various concentrations of Oroxylin A (T1 and T2) compared to T0. This seems to imply that the different concentrations of Oroxylin A could have resulted in different fine-tuning levels of the inhibition mechanism on *F. graminearum*, which, in turn, could have led to different levels of metabolites [25]. Notably, R5P was found be the only shared differential metabolite between both treatment concentrations (T1 and T2). It was reported that R5P is not only an important product of the pentose phosphate pathway [26], but can also be converted to phosphoribosyl pyrophosphate (PRPP) [27]. Speculatively, Oroxylin A suppresses the R5P concentration by possibly inhibiting the pentose phosphate pathway and, consequently, purine metabolism, by decreasing the supply of PRPP, which is required for ATP generation and purine nucleotide biosynthesis in *F. graminearum* (Figure 5B) [28,29]. Alternatively, a downregulated pentose phosphate pathway may result in decreased NADPH generation and increased oxidative stress, which imbalances the redox system in *F. graminearum*, thereby retarding its growth and development [30,31].

Most importantly, Oroxylin-A-treated *F. graminearum* had increased concentrations of *cis*-aconitate (T1) or fumarate (T2); both are important metabolic cycling products in the TCA cycle (Figure 5A) and were negatively correlated with *ZEB1* and *ZEB2* expressions. Therefore, the accumulation of *cis*-aconitate or fumarate suggests that the mitochondrial TCA cycle in *F. graminearum* might be inhibited, which would decrease ATP production, induce mitochondrial dysfunction, and reduce macromolecule (lipids, nucleotides, and proteins) biosynthesis. These metabolic imbalances would impair *F. graminearum* growth and ZEA production or even induce fungal death [32,33]. Moreover, the increased itaconic acid level in both T1 (FC = 1.75; *p* = 0.008) and T2 (FC = 1.51; *p* = 0.287) suggests that Oroxylin A may increase the decarboxylation of the TCA cycle’s intermediate *cis*-aconitate into itaconic acid, which is an important anti-bacterial compound [33].

## 4. Conclusions

To sum up, Oroxylin A effectively suppressed the growth of *F. graminearum* and ZEA production, and we suggest that Oroxylin A could be used as a promising antifungal and mycotoxin controlling agent.

## 5. Materials and Methods

### 5.1. Chemicals and Reagents

Oroxylin A (≥98% purity) and ZEA (>99% purity) standards were from Macklin Inc. (Macklin, Shanghai, China) and Sigma-Aldrich (St. Louis, MO, USA), respectively. Dimethyl sulfoxide (DMSO, 99.9%) was from Sigma-Aldrich. Methanol (Tedia, Fairfield, OH, USA), acetonitrile (Merck, Darmstadt, Germany), formic acid (>96%, Sigma-Aldrich), and water (Millipore, Bedford, MA, USA) were of HPLC grade. The fungal growth media, potato dextrose agar (PDA), and potato dextrose broth (PDB) were from Guangdong Huankai Microbial Sci. (Guangzhou, China).

### 5.2. Fungal Culture

The pathogenic ZEA-producing fungus, the *F. graminearum* strain (AF 2015031), isolated from diseased maize stems, was from the China Center for Type Culture Collection (CCTCC, Wuhan, China). Fungal cultures were maintained on PDA medium and were incubated in the dark at 28 °C, as described previously [16].

### 5.3. Analysis of Oroxylin A in PSE via UPLC-QTOF-MS/MS

#### 5.3.1. Preparation of Oroxylin A Standard Solution and PSE

Oroxylin A standard was made with up to 1 mg/mL stock solution in methanol. The stock solution was further diluted with methanol to produce a dilution series (20, 50, 100, 200, and 500 ng/mL), and was used to prepare calibration curves. PSE was prepared as described previously [16] and made up to 10 mg/mL in DMSO, then diluted to a final concentration of 1 mg/mL with methanol for quantitative analysis of Oroxylin A.

#### 5.3.2. UPLC-QTOF-MS/MS Analysis

Oroxylin A in PSE was quantified via UPLC-QTOF-MS/MS. Peak area was extracted and integrated after peak alignment and retention time correction. Calibration curves (five points) were obtained using external standard calibrations for Oroxylin A by injecting the standard solution in the concentration range of 20 ng/mL to 500 ng/mL. Calibration curves were established by plotting the peak area (y) versus concentration (x) of each analyte.

UPLC-QTOF-MS/MS analysis was performed using a Triple TOF^TM^ 5600 plus mass spectrometer (Sciex, Foster City, CA, USA) combined with an LC-30AD series ultra-performance liquid chromatography system (Nexera^TM^, Shimadzu, Tokyo, Japan). Chromatographic separations were achieved using a Zorbax Eclipse XDB-C18 column at 30 °C (2.1 × 100 mm × 3.5 μm, Agilent, Santa Clara, CA, USA) with a flow rate of 0.35 mL/min and a sample injection volume of 5 μL. The mobile phases were water/0.1% formic acid (A) and acetonitrile/0.1% formic acid (B). The linear gradient elution program was as follows: 40% B from 0 to 1 min; 100% B at 6 min; and 40% B from 6.01 to 11 min.

Samples were analyzed in the electrospray ionization (ESI)-negative mode. Data acquisition was carried out in full scan mode (*m*/*z* range from 70 to 1000) combined with information-dependent acquisition (IDA) mode. For IDA analysis, a collision energy of 30 eV was used to fragment the parent ions. The drying gas temperature was set at 450 °C (ESI-), and the ion spray voltage was −4500 V (ESI-). Atomization gas pressure, auxiliary heating gas pressure, and curtain gas pressure in negative ionization mode were set at 55, 55, and 35 psi, respectively. The instrument was mass calibrated automatically by infusing the Sciex Negative Calibration Solution (part no. 4460134) for negative mode after every six samples injected. One QC sample and one blank vial were run after every 10 samples.

### 5.4. Experimental Design

This study consisted of two experiments. Experiment Ⅰ was performed with antifungal assays, and Experiment Ⅱ was performed with a mycotoxin inhibition assay. For Experiment Ⅰ, PDA was used as the growth medium for fungal culture, and for Experiment Ⅱ, PDB liquid culture medium was used for fungal growth and mycotoxin production. Experiments Ⅰ and Ⅱ were both performed with 0 (control), 64, and 128 ng/mL Oroxylin A and designated as T0, T1, and T2, respectively. Three replicates of Experiment Ⅰ and five replicates of Experiment Ⅱ were performed. The Oroxylin A concentrations in T1 was equivalent to 1 mg of PSE, and that of T2 was equivalent to 2 mg of PSE, calculated based on the quantitation of Oroxylin A in PSE.

#### 5.4.1. Antifungal Activity of Oroxylin A against *F. graminearum*

The antifungal activity of Oroxylin A against *F. graminearum* was determined in vitro by measuring the linear growth rate of mycelium as reported previously [16]. Briefly, plugs of fungal growth on agar (4 mm diameter, from 7-day-old cultures) were inoculated onto plates and incubated at 28 °C for up to 5 days. The colony diameter of each plate was measured with the original mycelial plug diameter (4 mm) subtracted from the measurement after 3 and 5 days of inoculation, respectively, and the growth inhibition rate was calculated as follows: inhibition rate (%) = (Dc − Dt)/ Dc × 100%, where Dc represents the control (T0) colony diameter and Dt represents the colony diameter of the Oroxylin A treatments (T1 or T2). The results are presented as the mean ± standard error of the mean (SEM).

#### 5.4.2. Anti-Mycotoxin Activity of Oroxylin A against *F. graminearum*

The anti-mycotoxin activity of Oroxylin A in liquid cultures was determined as described previously [34] with minor modifications. In brief, a stock solution of Oroxylin A (1 mg/mL in DMSO) was prepared; then, an appropriate volume was added to a 100 mL centrifuge tube containing PDB (50 mL) to prepare media T1 and T2. Subsequently, fungal spore suspension (5 μL, 1 × 10^6^ spores/mL) from a 7-day-old culture was inoculated into tubes of T0, T1, and T2 media and inoculated with shaking (140–160 rpm) at 28 °C for 14 days.

After 14 days, the fungal biomass was separated from the culture medium with Whatman No. 1 filter paper, and the ZEA concentration in the filter was determined. The fungal mycelium was washed twice with sterile distilled water, transferred to cryotubes, frozen in liquid nitrogen, and stored at −80 °C until needed for RNA and metabolite extraction.

##### UPLC Analysis of ZEA

Before the UPLC analysis, the ZEA in the growth medium was extracted and cleaned up using an IAC-SEP^®^ ZEA-specific immunoaffinity column (Clover Technology Group, Beijing, China) according to the manufacturer’s instructions. The resulting sample was transferred to an autosampler vial, and 100 μL was injected into the UPLC column for analysis.

The ZEA content was tested following the Chinese standard method GB 5009.209-2016 [35]. Quantitative analysis was performed on a Dionex Ultimate 3000 UPLC system (Thermo Fisher Scientific, Waltham, MA, USA) consisting of an Ultimate 3000 RS autosampler, pump, column oven, and fluorescence detector, and controlled by Chromeleon 7.2 software. Separation was performed on a Zorbax Eclipse XDB-C18 RRHD column (100 × 2.1 mm, 1.8 μm; Agilent). The isocratic mobile phase was acetonitrile/water/methanol (46:46:8, *v*/*v*) with a flow rate of 1.0 mL/min. The fluorescence detection excitation wavelength was 274 nm, and the emission wavelength was 440 nm.

The ZEA was identified and quantified via comparison with the retention time and the chromatographic peak area of an external standard. The standard curve was obtained by preparing standard solutions at six ZEA concentrations (5, 10, 50, 100, 200, and 500 ng/mL), and each concentration was analyzed in triplicate. The resulting regression equation for ZEA was as follows: y = 2912.3x − 6081.6 (R^2^ = 0.9998).

##### mRNA Expression Analysis of ZEA Biosynthesis Genes via RT-PCR

The total RNA was extracted using Trizol reagent (Tiangen, Beijing, China), according to the manufacturer’s protocol. Quality and quantity of RNA were measured via RNA 6000 Nano LabChip kit using a Bioanalyzer (Agilent). The cDNA was synthesized using RevertAid Reverse Transcriptase following the manufacturer’s protocol (Thermo Fisher). Quantitative RT-PCR was performed in an ABI Q1 PCR system (Thermo Fisher) using PerfectStart Green qPCR SuperMix (TransGen, Beijing, China). In brief, the 10 μL reaction mixture consisted of 2 × SYBR Green RT-PCR reaction mix (5 μL), forward primer (10 nM; 0.3 μL), reverse primer (10 nM; 0.3 μL), cDNA (2 μL), and nuclease-free water (PCR grade; 2.4 μL). The temperature program comprised 10 min of starting template denaturation at 94 °C for 1 cycle, followed by 40 cycles of PCR at 95 °C for 5 s, 60 °C for 15 s, and 72 °C for 10 s. The primer sequences used were as described previously [36] and are listed in Appendix A. The synthesized primer sequences were obtained from Sangon Biotech (Shanghai, China). Elongation factor 1-α (*EF1-α*) was used as a house-keeping gene. Each reaction was completed with a melting curve analysis to ascertain that only the expected amplification products were generated. PCR data were analyzed using the 2^−ΔΔCT^ method to calculate the relative expression levels of the target gene.

##### Metabolomic Analysis

Fungal mycelia were extracted and analyzed as described previously [11]. Briefly, aliquots of mycelium (100 mg) were extracted with ice-cold methanol, and after centrifugation at 20,000× *g* for 10 min at 4 °C, the supernatant was collected, dried in an SPD121P SpeedVac vacuum concentrator (Thermo Fisher), and resuspended in acetonitrile. Analysis was performed via liquid chromatography/tandem mass spectrometry on a Triple TOF^TM^ 5600 plus mass spectrometer (Sciex) coupled to an LC-30AD series ultra-performance liquid chromatography system (Nexera^TM^, Shimadzu). The chromatographic separation was performed on an XBridge HILIC (2.1 × 100mm; 3.5 μm) column (Waters, Milford, MA, USA) and electrospray ionization (ESI) in positive and negative ion modes. The mobile phases were 50% acetonitrile/0.1% formic acid (A) and 95% acetonitrile/0.1% formic acid (B) at a flow rate of 0.25 mL/min. The linear gradient elution program was as follows: 80% B at 0 min, 20% B at 24 min, and 80% B at 24.5 min, held for 8.5 min.

The MS source parameters were as follows: drying gas temperature, 550 °C; spray voltage, 5000 V; and atomization gas, auxiliary heating gas, and curtain gas pressures of 45 psi, 45 psi, and 35 psi, respectively. Based on the original acquisition files, a pre-processing step with MarkerView 1.3.1 software (Sciex, Framingham, MA, USA) was performed for automated baseline correction and alignment of all extracted mass peaks across all samples.

Multivariate statistical analyses, including principal component analysis (PCA) and supervised partial least-squares discriminant analysis (PLS-DA) of the metabolic dataset, were performed using MetaboAnalyst 5.0 (http://www.metaboanalyst.ca/, assessed on 6 April 2023). Furthermore, the value of the fold change (FC) was calculated as the average normalized peak intensity ratio between samples. Differences between datasets with FC > 1.10 or FC < 0.90 and *p* < 0.05 (via Student’s *t*-test) were considered statistically significant.

### 5.5. Statistical Analysis

Pooled data are presented as the mean ± SEM. Comparisons among multiple samples were conducted via one-way analysis of variance (ANOVA) using SPSS 23.0 (IBM-SPSS Inc., Chicago, IL, USA). Tukey’s multiple comparison test was applied. Differences between groups were considered statistically significant at *p* < 0.05.

## Figures and Tables

**Figure 1 toxins-15-00535-f001:**
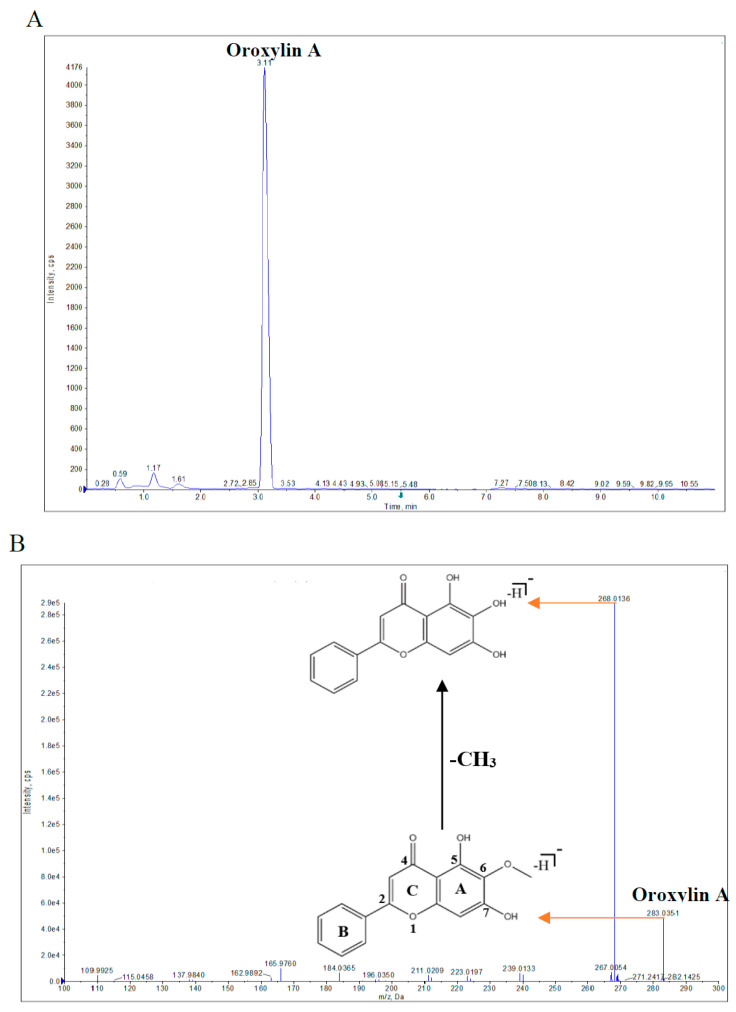
Extracted ion chromatogram (EIC) (**A**) and MS/MS spectrum (**B**) of Oroxylin A in PSE.

**Figure 2 toxins-15-00535-f002:**
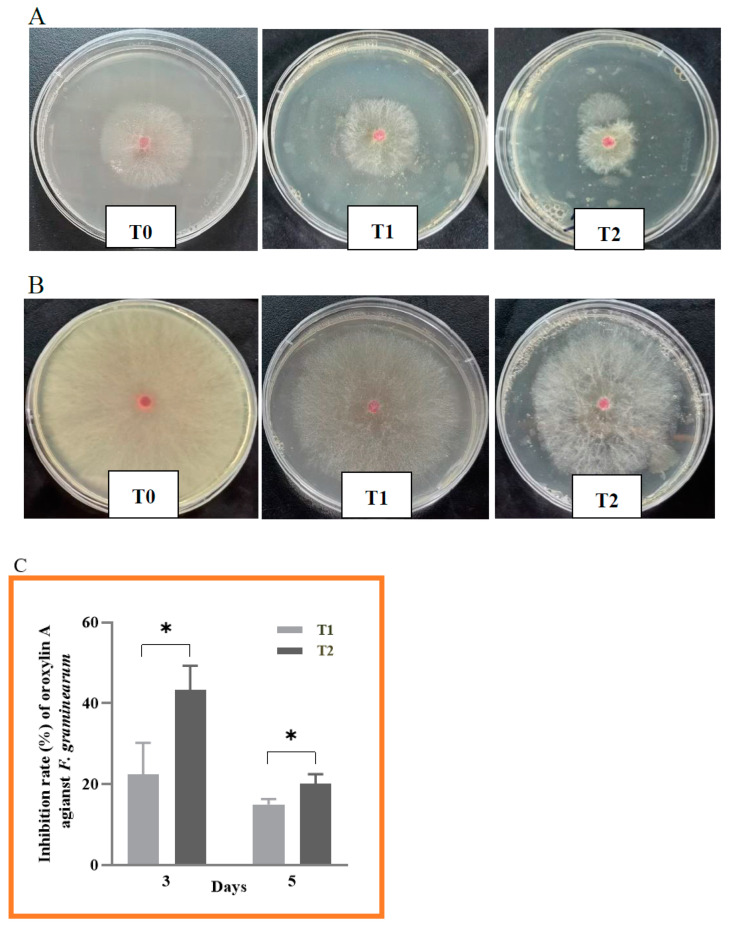
Effect of different concentrations of Oroxylin A on mycelial growth of *F. graminearum* 3 days (**A**) and 5 days (**B**) after inoculation. Oroxylin A inhibition of *F. graminearum* growth (**C**). All data are expressed as the mean ± SEM. Significance: * *p* < 0.05. T0-T2, means the three test concentrations of Oroxylin A used here were 0 (T0; control), 64 (T1), and 128 ng/mL (T2).

**Figure 3 toxins-15-00535-f003:**
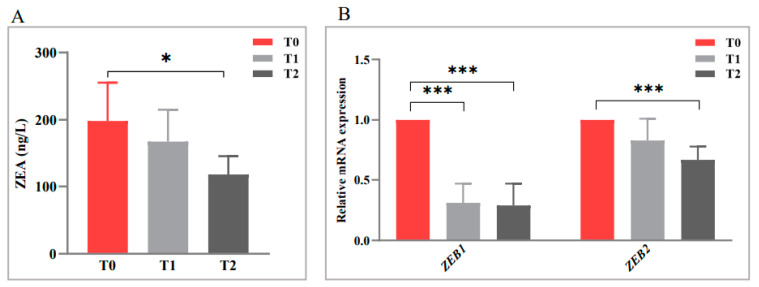
Inhibitory effect of Oroxylin A on (**A**) zearalenone (ZEA) production, and (**B**) the mRNA expression of ZEA-biosynthesis-related genes (*ZEB1* and *ZEB2*) of *F. graminearum* in liquid culture. All data are expressed as the mean ± SEM. Significance: * *p* < 0.05 and *** *p* < 0.001.

**Figure 4 toxins-15-00535-f004:**
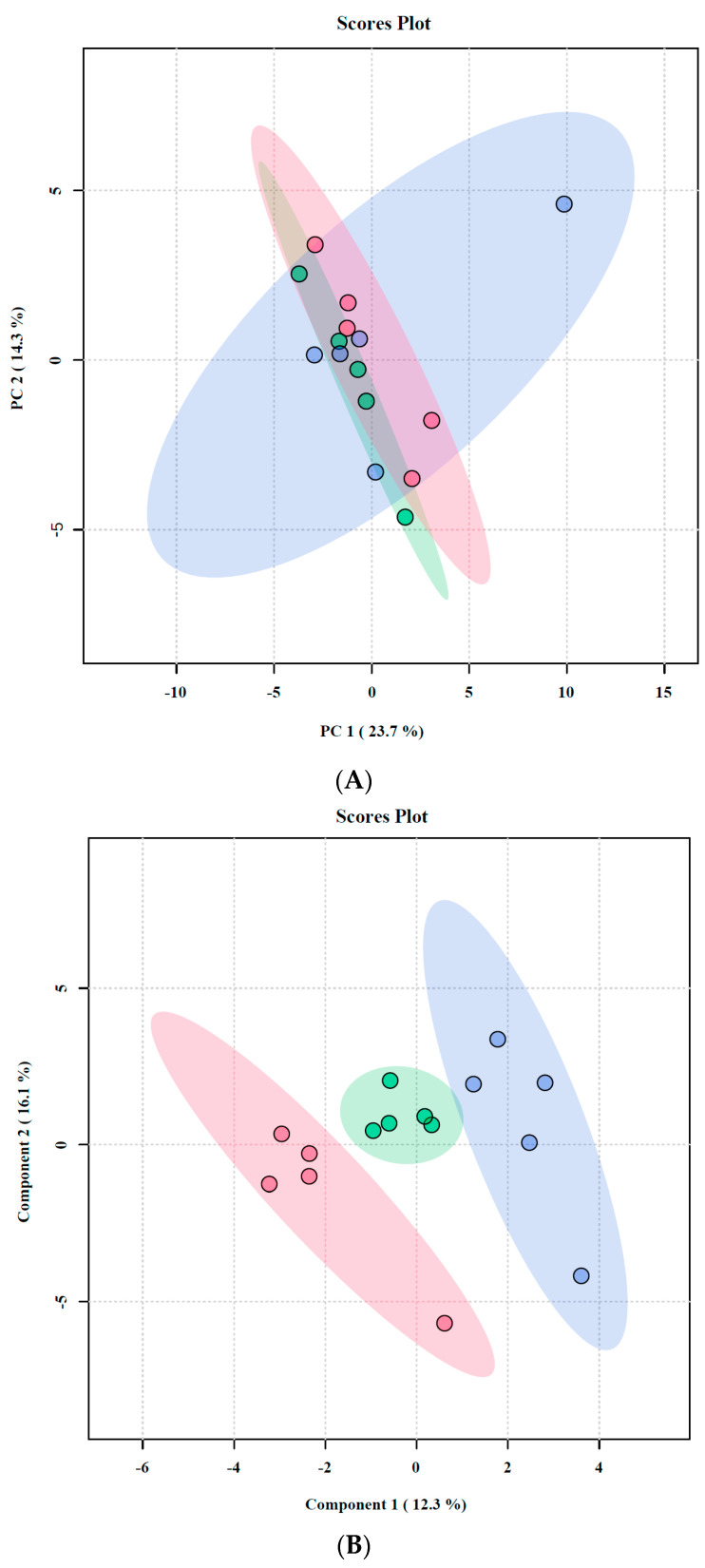
Metabolic changes in *F. graminearum* in response to Oroxylin A, comparing T1 vs. T0 and T2 vs. T0. (**A**) PCA analyses of metabolic datasets. (**B**) PLS-DA analyses of metabolic datasets. Symbols: T0 (red circle), T1 (green circle), and T2 (blue circle).

**Figure 5 toxins-15-00535-f005:**
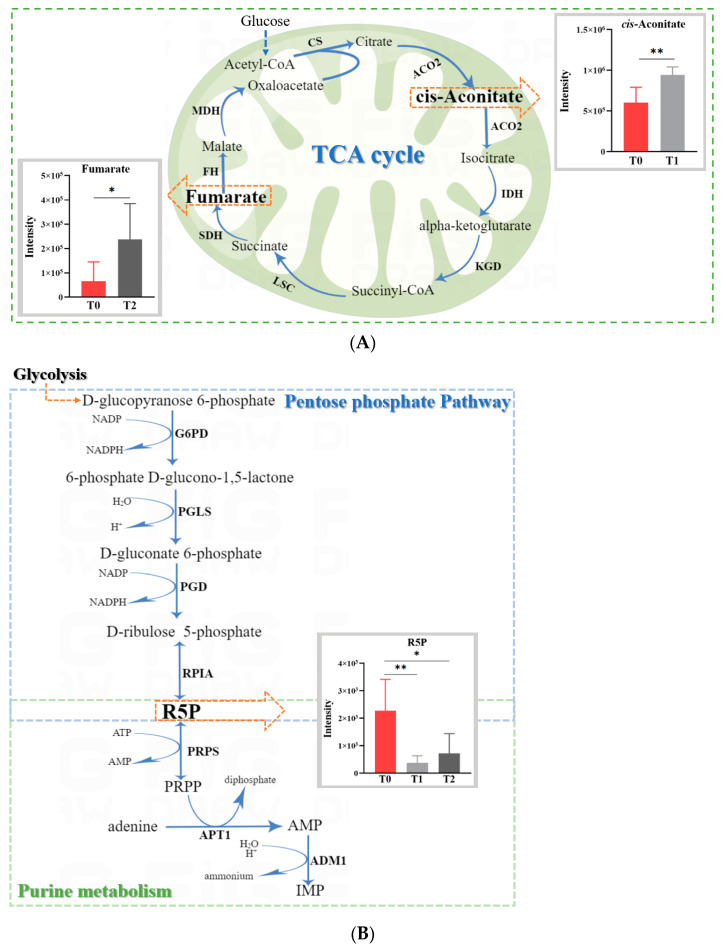
Common pathway analysis generated by MetaboAnalyst 5.0, based on differential metabolites from both T0 vs. T1 and T0 vs. T2. (**A**) TCA cycle; (**B**) pentose phosphate and purine metabolism pathways. For details in figures, refer to the SGD (http://pathway.yeastgenome.org/, assessed on 6 April 2023) and KEGG (http://www.genome.jp/kegg/, assessed on 6 April 2023) databases. Abbreviations: ACO2: aconitase; ADMI: adenosine 5-monophosphate deaminase; AMP: adenosine monophosphate; APTI: adenine phosphoribosyltransferase; CS: citrate synthase; FH: fumarate hydratase; G6PD: glucose-6-phosphate dehydrogenase; IDH: isocitrate dehydrogenase; IMP: inosine monophosphate; KGD: 2-ketoglutarate–dehydrogenase complex; LSC: succinyl-CoA ligase; MDH: malate dehydrogenase; PGD: 6-phosphogluconate dehydrogenase; PGLS: 6-phosphogluconolactonase; PRPP: 5-phospho-alpha-D-ribose 1-diphosphate; PRPS: ribose-phosphate pyrophosphokinase; RPLA: ribose 5-phosphate isomerase A; SDH: succinate dehydrogenase. All data are expressed as the mean ± SEM. Significance: * *p* < 0.05 and ** *p* < 0.01. Red color in figure means the differential metabolites.

**Figure 6 toxins-15-00535-f006:**
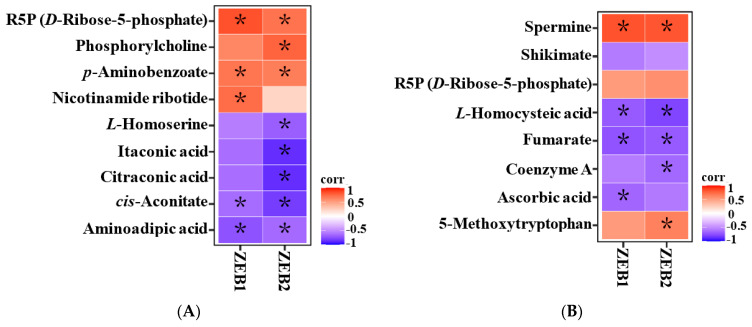
Pearson’s correlation analysis of differential metabolites and ZEA-biosynthesis-related genes between T1 and T0 (**A**), and between T2 and T0 (**B**). Each column represents a gene, each row represents a metabolite, and each square represents a correlation coefficient between the corresponding gene and metabolite. The color scale indicates the correlation strength, red indicates a positive correlation, and blue indicates a negative correlation. Significance: * *p* < 0.05.

## Data Availability

The data presented in this study are available upon request from the corresponding authors.

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
