# Peer review of "Antagonistic Activity of Oroxylin A against Fusarium graminearum and Its Inhibitory Effect on Zearalenone Production"

_toxins, 2023, doi:10.3390/toxins15090535_

Round 1

Reviewer 1 Report

The authors present a straight-forward study on the inhibitory effects of oroxylin A on the growth and zearalenone production of Fusarium graminearum. The manuscript is incomplete in its discussion. For example, the authors could give more attention to how the results are relevant to mycotoxin control. Both the abstract and conclusions sections restate the results without connecting them to wider implications (unlike how the authors decently interpret the biological involvements of differentially accumulated metabolites in lines 173-195).

The authors should report adjusted p-values for enriched KEGG pathways. Enrichment analyses involve multiple comparison testing. An introductory review on how and why p-values are adjusted is here: https://www.ncbi.nlm.nih.gov/pmc/articles/PMC6099145/. The organism used for KEGG reference pathways should be specified in the text.

Is PSE predominantly oroxylin A as suggested by Figure 1? Please clarify in the text.

Please state in the text why zearalenone was not listed as a differentially-accumulated metabolite in Figure 6 and supplementary tables.

Do the authors have an explanation for why T1 and T2 differentially-accumulated metabolites differ except for R5P and l-homocysteric acid, instead of observing an amplification of T1’s effects for T2? Please include the answer in the text.

The authors should provide a citation for “Chinese standard method GB 5009.209-2016”.

The journal will likely require better resolution for various figures.

For Figure S1, indicating negative intensities for the reference peaks is misleading. It would be better to have two separate plots or to remove/cover-up the negative signs on the y-axis.

There are minor grammatical, syntax and word usage issues such as inconsistent italicization of genus and species names, uppercase of oroxylin, word usage/conjugation (ex. line 5 “produce” -> “produces”, line 78 “exhibited” -> “inhibited”).

Author Response

1. The authors present a straight-forward study on the inhibitory effects of oroxylin A on the growth and zearalenone production of Fusarium graminearum. The manuscript is incomplete in its discussion. For example, the authors could give more attention to how the results are relevant to mycotoxin control. Both the abstract and conclusions sections restate the results without connecting them to wider implications (unlike how the authors decently interpret the biological involvements of differentially accumulated metabolites in lines 173-195).

We made appropriate modifications and refinements with red part in the abstract, conclusions and discussion sections, respectively.

2. The authors should report adjusted p-values for enriched KEGG pathways. Enrichment analyses involve multiple comparison testing. An introductory review on how and why p-values are adjusted is here: https://www.ncbi.nlm.nih.gov/pmc/articles/PMC6099145/. The organism used for KEGG reference pathways should be specified in the text.

We found that KEGG pathway enrichment analysis results did not meet criteria for corrected significance based on recalibrated P-values versus estimates based on the raw P-values. Nevertheless, this part of result does not influence the conclusion, so we removed them and made some modifications in the text with red font.

3. Is PSE predominantly oroxylin A as suggested by Figure 1? Please clarify in the text.

The details of our modifications are presented in the text with red font.

4. Please state in the text why zearalenone was not listed as a differentially-accumulated metabolite in Figure 6 and supplementary tables.

ZEA was present in filtered culture broth as described in Materials and methods, so ZEA production was measured from filtered culture broth by UPLC analysis (Figure 3A). While the remaining fungal mycelium after this filtration were harvested for metabolomics profiling (Figure 6 and Supplementary table S2) and genetic testing (Figure 3B).

5. Do the authors have an explanation for why T1 and T2 differentially-accumulated metabolites differ except for R5P and l-homocysteric acid, instead of observing an amplification of T1’s effects for T2? Please include the answer in the text.

The details of our modifications are presented in the text with red font. There are some similarities and differences of differential metabolites in the groups treated with various concentrations of Oroxylin A (T1 and T2) compared to T0. This seems to imply that the different concentration of Oroxylin A could have resulted in different fine turning of the inhibition mechanism on F. graminearum that in turn leaded to the different levels of metabolites

6. The authors should provide a citation for “Chinese standard method GB 5009.209-2016”.

A literature reference has been added about “Chinese standard method GB 5009.209-2016”. We added Ref. [36].

7.The journal will likely require better resolution for various figures.

High-resolution images of figures were provided. See Figure 4.

8. For Figure S1, indicating negative intensities for the reference peaks is misleading. It would be better to have two separate plots or to remove/cover-up the negative signs on the y-axis.

The negative signs on the y-axis were removed. See Figure S1.

9. There are minor grammatical, syntax and word usage issues such as inconsistent italicization of genus and species names, uppercase of oroxylin, word usage/conjugation (ex. line 5 “produce” -> “produces”, line 78 “exhibited” -> “inhibited”).

We agree with your suggestion and have modified them throughout the text as appropriate.

Reviewer 2 Report

Very well-written and presented manuscript.  

The only suggestion I have is to perhaps add the application of Oroxylin A within a commercial agriculture context, its implications etc. 

Author Response

1. The only suggestion I have is to perhaps add the application of Oroxylin A within a commercial agriculture context, its implications etc. 

We have supplemented the relevant information of Oroxylin A in the introduction section with red font.

Reviewer 3 Report

Dear Authors,

In this study, qualitative and quantitative analysis of oroxylin A in Piper sarmentosum extract and the antagonistic activity of oroxylin A against Fusarium graminearum and its zearalenone production.   For antifungal studies, a standard of oroxylin was used, not oroxylin from Piper sarmentosum extract. In the manuscript it does not understand other than in Material and Method section, so it is confusing.  The analysis of oroxylin A in Piper sarmentosum was previously published as cited by the authors. The manuscript may be arranged by removing qualitative and quantitative analysis of oroxylin A in Piper sarmentosum extract or it should be clarified in abstract, introduction and results and discussion. The abstract should also briefly describe the main methods or treatments and findings. Include the main methods and clarify the results.

For the introduction, according to the “ Instructions for Authors”, the introduction should briefly place the study in a broad context and highlight why it is important. Also, the current state of the research field should be reviewed carefully and key publications cited. There are different studies related to antimicrobial activity of Piper sarmentosum and oroxylin A in the literature. The general characteristics of oroxylin A and also its antimicrobial and other bioactivities should be summarized. The introduction should be rewritten according to the “ Instructions for Authors” and significantly improved.

In the manuscript, the results are generally presented in tables or figures and explained in detail. But the results are not discussed adequately by comparing the literature.  Also, the final conclusions should include key findings and more specific.

Generally, the language is fair enough and simple to understand. Still, some grammatical/ typographic/ punctuational errors need revision. 

Dear Editor, 

Generally, the language is fair enough and simple to understand. Still, some grammatical/ typographic/ punctuational errors need revision. 

Author Response

1. In this study, qualitative and quantitative analysis of oroxylin A in Piper sarmentosum extract and the antagonistic activity of oroxylin A against Fusarium graminearum and its zearalenone production.   For antifungal studies, a standard of oroxylin was used, not oroxylin from Piper sarmentosum extract. In the manuscript it does not understand other than in Material and Method section, so it is confusing.  The analysis of oroxylin A in Piper sarmentosum was previously published as cited by the authors. The manuscript may be arranged by removing qualitative and quantitative analysis of oroxylin A in Piper sarmentosum extract or it should be clarified in abstract, introduction and results and discussion. The abstract should also briefly describe the main methods or treatments and findings. Include the main methods and clarify the results.

Our previous study found that PSE had antifungal activity against F. graminearum and 17 compounds were tentatively identified as the main antifungal constituents of PSE. Among the 17 compounds, Oroxylin A is the only flavonoid, and this was the first report of Oroxylin A in P. sarmentosum. Although Oroxylin A is one of the main antifungal compounds of PSE, we merely performed the tentatively identification of Oroxylin A in PSE through the mzCloud database (https://www.mzcloud.org) in our previous study. Therefore, in the present study, UPLC-QTOF-MS was used for further qualitative and quantitative validation of Oroxylin A in PSE based on the MS and MS2 fragmentation patterns of a Oroxylin A reference standard, in order to enable determination of the suitable dosages of Oroxylin A for subsequent experiments. We made minor modifications and supplementations and marked with red font in the Introduction, Material and Method sections. We also clarified qualitative and quantitative analysis of oroxylin A in P. sarmentosum extract in abstract, introduction and results and discussion.

2. For the introduction, according to the “ Instructions for Authors”, the introduction should briefly place the study in a broad context and highlight why it is important. Also, the current state of the research field should be reviewed carefully and key publications cited. There are different studies related to antimicrobial activity of Piper sarmentosum and oroxylin A in the literature. The general characteristics of oroxylin A and also its antimicrobial and other bioactivities should be summarized. The introduction should be rewritten according to the “ Instructions for Authors” and significantly improved.

The details of our modifications are presented in the introduction section with red font.

3. In the manuscript, the results are generally presented in tables or figures and explained in detail. But the results are not discussed adequately by comparing the literature.  Also, the final conclusions should include key findings and more specific.

We made appropriate modifications in the discussion and conclusion sections.

4. Generally, the language is fair enough and simple to understand. Still, some grammatical/ typographic/ punctuational errors need revision. 

We agree with your suggestion and have modified them throughout the text as appropriate.

Round 2

Reviewer 3 Report

The manuscript is acceptable in its present form.